# Comparing multiscale, presence-only habitat suitability models created with structured survey data and community science data for a rare warbler species at the southern range margin

**Lauren E. Whitenack**[1,2]*, **Sara J. Snell Taylor**[1], **Aimee Tomcho**[3], **Allen H. Hurlbert**[1,4]

1 Department of Biology, University of North Carolina, Chapel Hill, NC, United States of America,
2 Department of Biology and Ecology, Evolution, and Conservation Biology Graduate Program, University of Nevada, Reno, NV, United States of America, 3 Audubon North Carolina, Durham, NC, United States of America, 4 Environment, Ecology and Energy Program, University of North Carolina, Chapel Hill, NC, United States of America

* lauren.whitenack@gmail.com

**Data Availability Statement:** The authors did not collect and do not own the data underlying the

## Abstract

Golden-winged Warblers (*Vermivora chrysoptera*, Parulidae) are declining migrant songbirds that breed in the Great Lakes and Appalachian regions of North America. Within their breeding range, Golden-winged Warblers are found in early successional habitats adjacent to mature hardwood forest, and previous work has found that Golden-winged Warbler habitat preferences are scale-dependent. Golden-winged Warbler Working Group management recommendations were written to apply to large regions of the breeding range, but there may be localized differences in both habitat availability and preferences. Rapid declines at the southernmost extent of their breeding range in Western North Carolina necessitate investigation into landscape characteristics governing distribution in this subregion. Furthermore, with the increase in availability of community science data from platforms such as eBird, it would be valuable to know if community science data produces similar distribution models as systemic sampling data. In this study, we described patterns of Golden-winged Warbler presence in Western North Carolina by examining habitat variables at multiple spatial scales using data from standardized Audubon North Carolina (NC) playback surveys and community science data from eBird. We compared model performance and predictions between Audubon NC and eBird models and found that Golden-winged Warbler presence is associated with sites which, at a local scale (150m), have less mature forest, more young forest, more herb/shrub cover, and more road cover, and at a landscape scale (2500m), have less herb/shrub cover. Golden-winged Warbler presence is also associated with higher elevations and smaller slopes. eBird and Audubon models had similar variable importance values, response curves, and overall performance. Based on variable importance values, elevation, mature forest at the local scale, and road cover at the local scale are the primary variables driving the difference between Golden-winged Warbler breeding sites and random background sites in Western North Carolina. Additionally, our results validate the use of

results presented in the study. Environmental raster data are available from the following sources: LANDFIRE data from https://landfire.gov; Rangeland Analysis Platform data from http://rangeland.ntsg.umt.edu/data/rap/rap-vegetation-cover/; National Landcover Dataset Canopy Cover data from https://www.mrlc.gov/data; and ASTER Digital Elevation Model data from https://earthexplorer.usgs.gov/. eBird data are available for download at ebird.org. Audubon North Carolina data can be accessed by contacting Curtis Smalling, Director of Conservation at Audubon North Carolina, via email at Curtis.Smalling@audubon.org. The authors accessed Audubon North Carolina data by email request and received no special access privileges that others would not have.

**Funding:** The authors received no specific funding for this work.

**Competing interests:** The authors have declared that no competing interests exist.

eBird data, since they produce species distribution modeling results that are similar to results obtained from more standardized survey methods.

## Introduction

Habitat loss is a primary threat to biodiversity in the present day [1, 2]. Migratory birds may be especially vulnerable to habitat loss since they rely on the persistence of multiple quality habitats for breeding, stopover, and wintering [3, 4]. As such, understanding the habitat associations of migratory birds can help predict patterns of distribution and abundance, and inform management practices and conservation efforts. The Golden-winged Warbler (*Vermivora chrysoptera*, Parulidae) is a migrant songbird which has been declining at an average of 1.85% per year for over 50 years across their range [5]. Due to its vulnerable status, the Golden-winged Warbler has been assigned to the Partners in Flight Red Watch List [6], the United States Fish and Wildlife Service's list of Birds of Conservation Concern [7] and is a candidate for listing under the Endangered Species Act [8].

Currently, Golden-winged Warblers breed in two main regions: the Great Lakes region of southeastern Canada and the northern-midwestern United States, and the Appalachian region in select moderate-to-high elevation sites in the Appalachian Mountains [9]. Within their breeding range, Golden-winged Warblers are generally found in early successional habitats near mature hardwood forest [10, 11]. During the breeding season, Golden-winged Warblers use multiple vegetation layers. Golden-winged Warblers build their nests on the ground in the herbaceous layer or just above the ground in shrubs [12, 13]. Shrub cover provides protection from predators and surrounding hardwood forest is used for male perches, nesting material, foraging ground, and post-fledging habitat [11, 12, 14–16]. Many bird species, including Golden-winged Warblers, select habitat based on conditions at multiple spatial scales, narrowing down potential sites from large to small scales [17–20]. Studies of Golden-winged Warbler breeding habitat associations have shown that variables important at the scale of the nest site, such as herbaceous and shrub cover, differ from variables important at larger scales, such as mature forest [21–23].

Despite the general trends, Golden-winged Warbler breeding habitat associations vary between geographical areas and with landscape context [21–23]. Since Golden-winged Warblers have a large latitudinal breeding range spanning from Canada to Georgia, both the availability of certain habitat characteristics and preferences for different habitat characteristics (local adaptation or behavioral plasticity) may explain this variation in habitat associations [24]. Thus, it is important to study Golden-winged Warbler breeding habitat associations throughout their breeding range and at multiple spatial scales to understand these regional differences. Many Golden-winged Warbler habitat studies are conducted in the Great Lakes Region and the central Appalachians where densities are high [10, 11, 16, 21, 25]. Understanding Golden-winged Warbler habitat associations in less-studied parts of their range is therefore a priority.

Golden-winged Warblers are especially vulnerable in the southern Appalachian Mountains at the southernmost extent of their breeding range. Data from the North American Breeding Bird Survey indicate that in Western North Carolina, Golden-winged Warbler populations have decreased by approximately 6.5% per year from 1993–2019 [5]. Habitat loss due to human development and maturation of early successional habitat, as well as brood parasitism by Brown-headed Cowbirds (*Molothrus ater*) have contributed to this decline [12]. Thus, declines in Golden-winged Warbler populations necessitate further investigation into the habitat associations of this species, especially at the southern limit of its breeding range.

The Golden-winged Warbler Working Group (GWWG) was founded in 2003 to facilitate collaboration among scientists to produce best management practices for the conservation of Golden-winged Warblers throughout their breeding and wintering ranges [26]. The GWWG best management practices for the Appalachian Region are designed to apply to Golden-winged Warbler populations in the Appalachian Mountains from New York to Georgia. Because local Golden-winged Warbler habitat associations may vary within the large latitudinal gradient of the Appalachian Region, it is critical to examine how these management guidelines align with Golden-winged Warbler habitat associations across the region. Difficulty in identifying early successional habitat with appropriate granularity across large spatial scales has prevented large-scale quantitative analyses of habitat associations to support these recommendations. Fine-tuning these management recommendations based on localized differences in habitat associations could vastly improve conservation efforts of Golden-winged Warblers, especially at the limits of their breeding range.

For over thirty years, Audubon North Carolina (Audubon NC) has conducted playback surveys during breeding season to collect data on the abundance and distribution of breeding Golden-winged Warblers in Western North Carolina using the Golden-winged Warbler Atlas Project protocol [27]. Audubon NC survey locations are chosen based on drive-by habitat assessments, aerial photo review, proximity to known locations within dispersal range, and private landowner cooperation, which could focus survey effort on certain parts of the landscape while excluding others. Audubon NC surveys are conducted with the primary goal of finding new Golden-winged Warbler habitat, and much of what is known about the current distribution of Golden-winged Warblers in North Carolina can be attributed to Audubon NC and the Golden-winged Warbler Atlas Project. Analysis of the habitat associations of breeding Golden-winged Warblers in Western North Carolina could help identify parts of the landscape that may be suitable for breeding birds but have not been surveyed.

With the increase in popularity of the community science platform eBird (ebird.org), avian presence and abundance data is now freely available for scientists to use to study species' ranges and track changes in distribution over time [28]. eBird users can submit bird observations at any location and time, resulting in over 70 million complete checklists worldwide, and over 1 million in North Carolina alone at the time of this publication [29]. Notably, eBird data are considered semi-structured and are usually collected by non-professionals, potentially resulting in a noisier dataset [30]. Unlike structured surveys such as the Golden-winged Warbler Atlas Project, eBird data are not usually collected by observers with a specific conservation or scientific goal. Despite these shortcomings, eBird data are increasingly being used successfully to understand distributions and habitat associations of bird species [31–35].

Community science data require fewer resources to collect than more traditional survey methods, which require time and financial resources to organize and implement. Since Golden-winged Warblers are rare in the Western North Carolina subregion, extra effort is required to locate breeding sites. Many structured survey methods involve the use of playback, in which a series of conspecific and/or allospecific bird songs or calls are broadcast to elicit a response from a target species. While some eBird users may use a minimal amount of playback, most are likely only observing, and long periods of playback use are not part of the eBird data collection method. By contrast, Audubon NC surveys use both conspecific and predator sounds in a 20-minute-long standard playback protocol [27]. While there is limited evidence of detrimental effects of conspecific playback on songbirds [36], there is substantial evidence of reduced reproductive output with increased perceived predation risk from predator playback [37]. Importantly, Audubon NC playback surveys are conducted at maximum once per year per site, minimizing such detrimental effects. Thus, the main differences between Audubon NC and eBird data include: 1) Audubon NC surveys are conducted by trained staff or

volunteers for targeted conservation work whereas eBird data are mostly collected by non-professionals without specific conservation or management goals; and 2) Audubon NC surveys involve the use of a standardized playback protocol, whereas eBird data do not usually involve the use of playback, and are not collected following a survey protocol. Since structured surveys require time and financial resources, and involve the use of playback, it would be valuable to know if Golden-winged Warbler habitat models created with eBird community science data produce similar results to those created with such structured survey data.

Because both structured survey data (Audubon NC) and eBird community science data are abundant and available in the area, Western North Carolina is the ideal study site to compare habitat models created with these two datasets. Additionally, Western North Carolina is a sub-region of considerable conservation concern, since it is located at the southernmost extent of the Golden-winged Warbler breeding range. Our goals in this paper are twofold: (1) to describe habitat associations of Golden-winged Warblers in Western North Carolina at multiple spatial scales and compare these associations to other areas within the breeding range, and (2) to determine whether a model created using eBird data would yield the same results as a model created with Audubon NC data.

## Methods

We studied Golden-winged Warbler habitat associations across Western North Carolina within the breeding range defined by the U.S. Geological Survey–Gap Analysis Project [38] (Fig 1).

We conducted our analyses using two sets of Golden-winged Warbler presence data: Audubon NC survey data and community science data from eBird. First, we obtained Golden-winged Warbler presence data from Audubon NC collected during breeding season (May-July) from 2000–2020. The data from Audubon NC were collected using standardized playback surveys as outlined in the Golden-winged Warbler Field Survey Protocol prepared by the Cornell Lab of Ornithology [27]. Audubon NC surveys were conducted at sites that were determined by visual inspection to be potentially suitable, usually roadside or on private lands managed for Golden-winged Warblers with the permission of the landowner. Second, we downloaded Golden-winged Warbler observations from eBird during the breeding season from 2000–2020 [29]. We used stationary and incidental checklists only, excluding traveling checklists for which there is greater spatial uncertainty surrounding the precise location of target birds.

Notably, the Audubon NC dataset contains Golden-winged Warbler absences and absences can be inferred from complete eBird checklists [30]. We decided not to use absences in our analysis for several reasons: 1) Absences are not comparable across datasets because all Audubon absences are in locations pre-determined to be potentially suitable for Golden-winged Warblers, while eBird-inferred absences are not; and 2) Audubon surveys included the use of standardized playback, which increases detection probability for this rare species, while eBird surveys did not, thus increasing the likelihood of false absences in the eBird dataset [39]. Because we chose not to use absences in our analysis, we employed a Maxent modeling approach, which has been shown to perform better than generalized linear modeling and other methods when presence-background data are used [40].

We removed eBird checklists that were submitted under the Golden-winged Warbler Atlas Project protocol, since Audubon NC now submits their survey results to eBird. To further correct for data redundancy, we searched both datasets for all presence locations within 100m of each location and kept only the location of the most recent Golden-winged Warbler observation. We chose this distance because 100m is the low end of the maximum detection distance

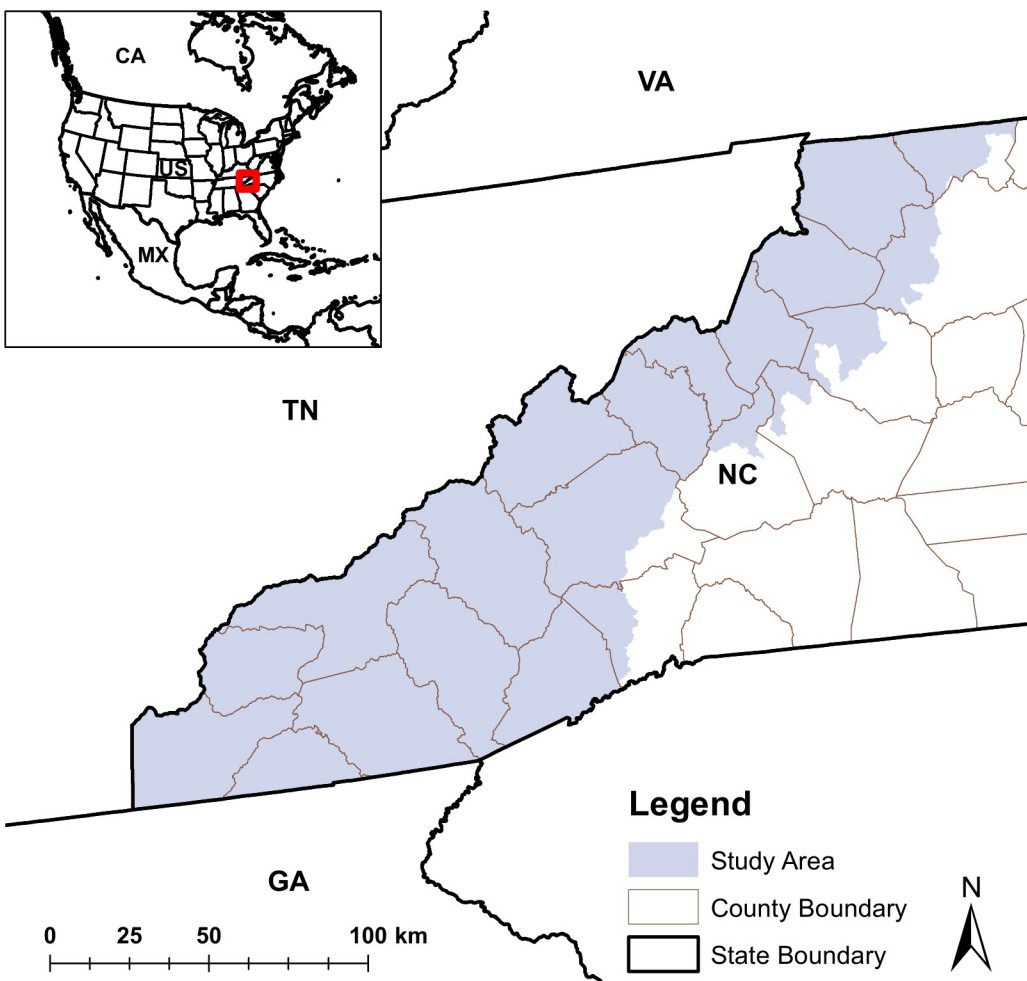

**Fig 1. Study area in Western North Carolina, United States.** Study area boundary was created using the U.S. Geological Survey–Gap Analysis Project Golden-winged Warbler breeding range [38].

for mixed shrub and forest habitats and is the approximate median radius of territory size identified in the GWWG conservation plan for management purposes [26, 39]. Since some territories were occupied in multiple years but the recorded spatial locations may differ across years, this redundancy analysis allowed us to eliminate duplicate points from the same territories. Notably, many known Golden-winged Warbler locations visited by eBirders may have been initially identified by Golden-winged Warbler Atlas Project surveys, and vice versa. To account for this overlap between datasets, we assigned each point to the dataset with the earliest record of a presence within 100m of that point, starting in 1988 when the Audubon NC surveys began. This redundancy analysis resulted in an Audubon sample size of N = 279 presence points and an eBird sample size of N = 86 presence points.

We obtained raster data from four different data sources to create landcover and topographic covariates that were added into our habitat models. First, we obtained landcover data from the United States Forest Service LandFire Data Distribution Site with a pixel size of 30x30m [41–46]. We used the LandFire Existing Vegetation Height (EVH) dataset because unlike other landcover datasets, it separates forest into different height categories, allowing us to distinguish young forest from mature forest. For forest cover variables, we used only

LandFire EVH data from 2008, 2012, and 2014 because the other available datasets (2001, 2016, and 2020) employed different methods and categorization schemes for forest height classification and are not consistent across time in our study area. For other landcover variables that we derived from LandFire EVH (roads, developed land, agricultural land), we used all available years. Next, we used the National Landcover Dataset (NLCD) United States Geological Survey (USGS) Canopy Cover data from 2011 and 2016, which are continuous raster datasets describing percent canopy cover within each 30x30m pixel [47]. Third, we used the Rangeland Analysis Platform (RAP) vegetation cover data, which describes percent herbaceous and shrub cover within each 30x30m pixel [48]. One aim of the RAP project is to create landcover products that more accurately describe herbaceous and shrub cover since other categorical landcover datasets often underrepresent these early successional landcover types. RAP data were available for every year represented in our Golden-winged Warbler presence datasets, so we downloaded data for all years from 2000–2020. Finally, we used the Advanced Spaceborne Thermal Emission and Reflection Radiometer (ASTER) Global Digital Elevation Model (DEM) Version 3 for our topographic variables, which has a 30x30m resolution [49]. For a summary of raster data used in our analysis, please see S1 Table.

Because Golden-winged Warblers nest in early successional habitat but also prefer surrounding mature forest, we investigated habitat associations at two spatial scales: the local scale (within 150m) and the landscape scale (within 2500m). We chose 150m because we aimed to identify early successional habitat patches at a relatively small scale that still accounted for some spatial error inherent in both datasets (playback response distance in the Audubon NC dataset and known potential spatial noise in the eBird dataset [30]). We chose the 2500m buffer distance to capture landscape-scale patterns because this distance is used in the GWWG management guidelines as well as other Golden-winged Warbler studies [22, 26].

Using the *raster* package in R, we created two sets of rasters from each of the initial Land-Fire datasets: one set describing percent land cover within a 150m circular buffer of each pixel and the other describing percent land cover within a 2500m circular buffer of each pixel [50, 51]. Layers describing percent land cover within a 150m buffer included percent forest of height 0–10 meters, percent forest of height 25-50m, and percent road cover. Layers describing percent land cover type within a 2500m buffer included percent forest of height 25-50m, percent road cover, percent agricultural land, and percent developed land. For each of the two buffer distances (150m and 2500m), we also calculated percent canopy cover using the NLCD USFS Canopy Cover datasets and percent herb/shrub cover using the RAP datasets. We used the ASTER DEM dataset to calculate slope and aspect according to Horn (1981) with the *raster* package in R [51, 52].

We ran separate models for Audubon NC and eBird datasets using the same set of background points. We created 10,000 background points by sampling from polygons that extended 10km around each presence point in a combined Audubon/eBird dataset using the *dismo* and *sp* packages in R [50, 53–55]. We extracted environmental variables at presence (Audubon NC, N = 279; eBird, N = 86) and background points in each dataset. For presence points, we extracted landcover values from the raster dataset with the closest year to the observation date. For background points, we extracted landcover values from the most recent available dataset. Since topographic data are more consistent over time, we used 2019 topographic data for all presence and background points.

Before modeling, we performed a Spearman's correlation analysis between all extracted variables in the presence and background datasets. Variables with a correlation coefficient >0.8 in any of the datasets were not included in the same model [56, 57]. Since canopy cover and herb/shrub variables were highly correlated (>0.8) at both spatial scales, and the RAP data have a finer temporal resolution than the NLCD Canopy Cover data, we decided to include herb/shrub variables and exclude canopy cover variables in our models. Road cover and

developed land were highly correlated (>0.8) at the landscape scale (2500m), so we used only the developed land variable and excluded the road cover variable at the landscape scale.

We used a Maxent modeling approach starting with model tuning using the *ENMeval* package in R [58, 59]. Using the function ENMevaluate, we compared Maxent models created with different combinations of feature classes and regularization multipliers. We excluded the "product" and "threshold" feature classes from our combinations of tuning arguments based on our expectations for the shapes of responses to landcover and topographic variables [60]. We also excluded the "threshold" feature class because it requires 80 presence records for training and our eBird dataset had only 86 presence points (not leaving enough points for cross-validation) [61]. This left us to compare models with "linear", "quadratic", and "hinge" feature classes, and regularization multipliers of 0.5, 0.75, 1, 1.25, 1.5, 1.75, 2, 5, 10, 15, and 20 [62]. To identify the best tuning arguments for our models, we used 10-fold cross-validation where we set aside 10% of the data for testing and repeated the analysis 10 times until each presence point was used for both training and testing. We used these cross-validation results to select the model with the lowest average test omission rate [59, 63]. Since we wanted to create models that were comparable across Audubon and eBird datasets, we selected the same tuning parameters for both models by identifying the tuning parameters that led to the lowest average test omission rate for both models. This tuning process led us to select the combination of "linear", "quadratic" and "hinge" feature classes and a regularization multiplier of 15. We calculated the area under the receiver operating characteristic curve (AUC) using the ENMevaluate function to assess model performance.

Using the ENMnulls function from the *ENMeval* package in R, we ran null simulations with 100 iterations to obtain a distribution of model performance metrics from null models [59, 64, 65]. We compared null performance (AUC) with empirical model performance to determine whether our models performed better than what would be expected from a null model.

We created response curves from the Maxent model output showing the probability of Golden-winged Warbler occurrence for varying values of each contributing variable (Fig 2). We predicted current suitable habitat across the landscape of the study area based on our models using the predict function from the *dismo* package and the most recent landcover data (Fig 3) [53]. To test how well our models created with one dataset performed when predicting the other dataset, we extracted predicted values at each presence and background point of the other dataset (extracted Audubon prediction values for all eBird points, and vice versa). We used the ROCR package in R to calculate AUC for these cross-dataset predictions.

## Results

Model results from the Audubon NC dataset indicate that at the local scale (within 150m) Golden-winged Warbler presence was positively associated with forest of height 0-10m, herb/shrub cover, and road cover, and negatively associated with forest of height 25-50m (Fig 2). At the landscape scale (within 2500m), presence was negatively associated with developed land cover and herb/shrub cover (Fig 2). Probability of occurrence decreased slightly when proportion of forest of height 25-50m within 2500m was greater than 0.3 (Fig 2). Variables that contributed the most to the Audubon model include elevation (34.6%), forest of height 25-50m within 150m (26.2%), developed land within 2500m (14.5%), and road cover within 150m (13.1%) (Table 1). Variables that did not contribute to the Audubon model include agricultural land within 2500m (Table 1). Notably, aspect and forest of height 25–50 within 2500m contributed very little to the Audubon model (Table 1). The Audubon model received an average test

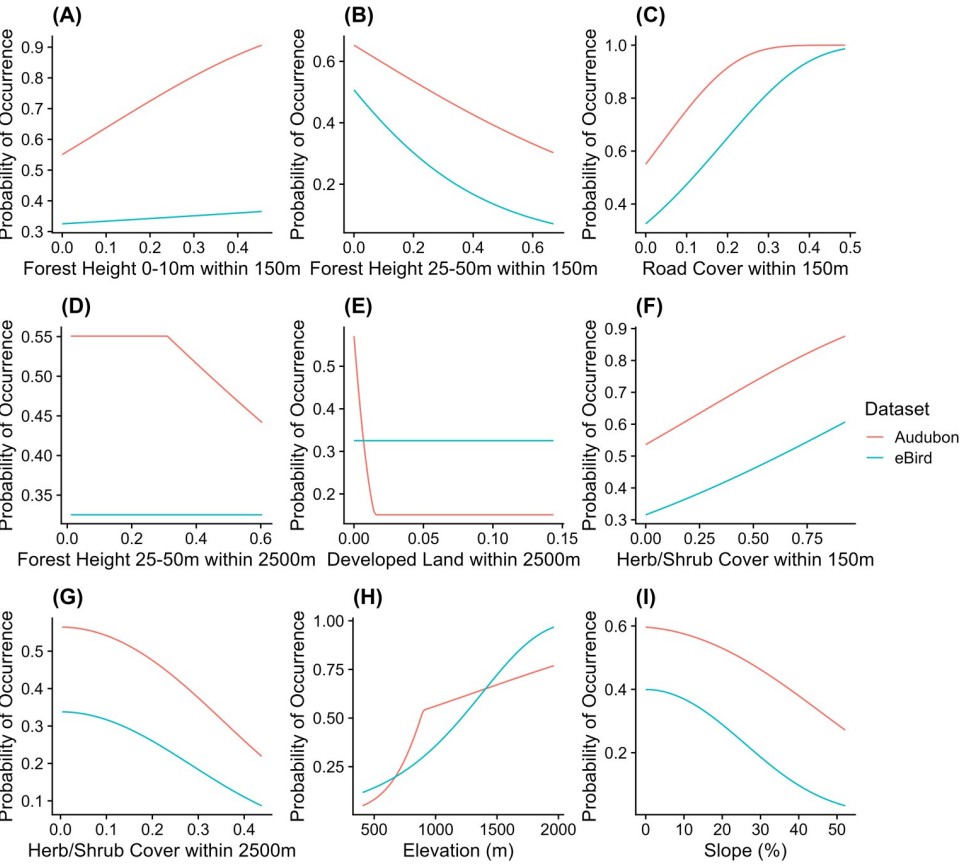

**Fig 2. Response curves of contributing variables from Maxent models created with Audubon and eBird datasets.**

AUC value of 0.80 ± 0.06 and performed significantly better than the null (average test AUC, P < 2e-16).

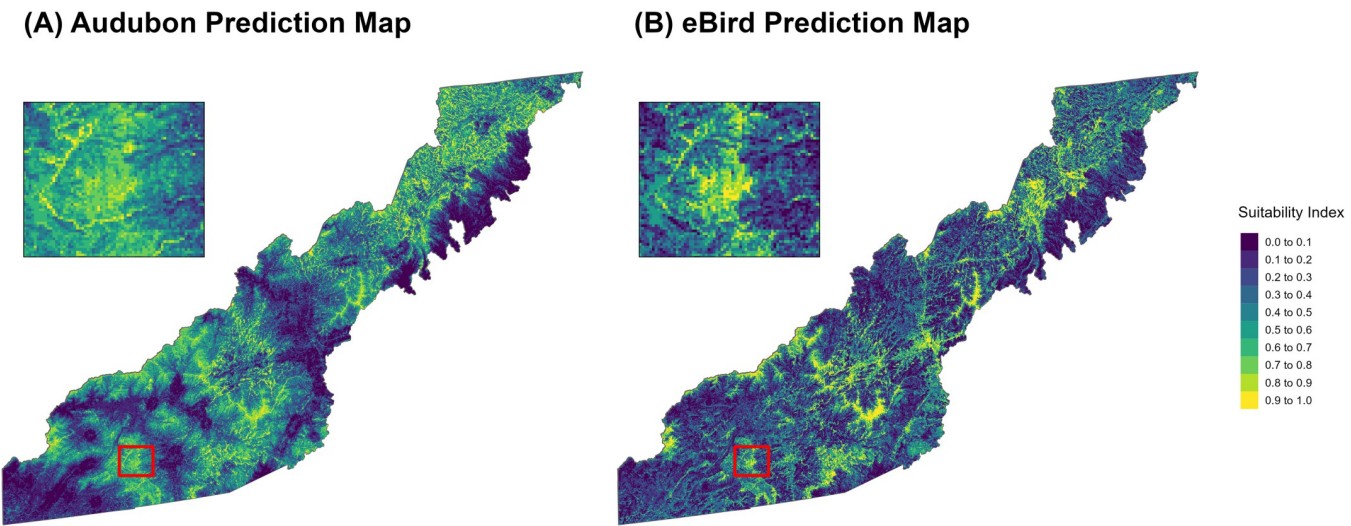

**Fig 3.** Habitat suitability prediction maps created from (A) Audubon and (B) eBird models.

**Table 1. Variable importance values from Maxent models created with Audubon and eBird datasets.**

| Variable | Audubon model | | eBird model | |
|---|---|---|---|---|
| | Percent Contribution | Permutation Importance | Percent Contribution | Permutation Importance |
| Forest height 0-10m within 150m | 1.4 | 1.4 | 0.01 | 0.07 |
| Forest height 25-50m within 150m | 26.2 | 0.9 | 49.0 | 42.9 |
| Road cover within 150m | 13.1 | 19.3 | 14.9 | 15.0 |
| Forest height 25-50m within 2500m | 0.2 | 1.2 | 0 | 0 |
| Agricultural land within 2500m | 0 | 0 | 0 | 0 |
| Developed land within 2500m | 14.5 | 19.3 | 0 | 0 |
| Herb and shrub cover within 150m | 6.7 | 2.9 | 1.7 | 4.3 |
| Herb and shrub cover within 2500m | 1.5 | 4.7 | 2.3 | 5.6 |
| Elevation | 34.6 | 48.0 | 25.5 | 19.9 |
| Slope | 1.7 | 2.3 | 6.6 | 12.3 |
| Aspect | 0.009 | 0.05 | 0 | 0 |

eBird model results were remarkably similar to Audubon model results. Based on the eBird model, at the local scale, presence was positively associated with percent forest of height 0-10m, herb/shrub cover, and road cover, and negatively associated with percent forest of height 25-50m (Fig 2). At the landscape scale (within 2500m), presence was negatively associated with herb/shrub cover (Fig 2). Variables that contributed the most to the eBird model include forest of height 25-50m within 150m (49.0%), elevation (25.5%), and road cover within 150m (14.9%) (Table 1). Variables that did not contribute to the eBird model include forest of height 25-50m within 2500m, agricultural land within 2500m, developed land within 2500m, and aspect (Table 1). Notably, forest of height 0-10m within 150m contributed very little to the eBird model (Table 1). The eBird model received an average test AUC value of 0.81 ± 0.08 and performed significantly better than the null (average test AUC, $P < 2e\text{-}16$). Means and standard deviations of all habitat variables for Audubon, eBird, and background data points are included in S2 Table.

Habitat suitability maps show that most of our study area is not well-suited for breeding Golden-winged Warblers based on the models (Fig 3). Mean suitability values across the landscape of the study area were 0.40 ± 0.23 based on the Audubon model and 0.35 ± 0.24 based on the eBird model. The eBird and Audubon prediction rasters were positively correlated (Spearman's correlation coefficient = 0.66, $P < 2e\text{-}16$). Both predictions were sufficient at differentiating between presence and background points of the other dataset (eBird prediction values and Audubon presence points, AUC = 0.72; Audubon prediction values and eBird presence points, AUC = 0.81).

## Discussion

We predicted habitat suitability across Western North Carolina using community science data from eBird and structured survey data from Audubon NC. Our results suggest that in Western North Carolina, Golden-winged Warblers are found at sites with less mature forest, more young forest, more herb/shrub cover, and more road cover at a local scale (within 150m, Fig 2). At a landscape scale (within 2500m), Golden-winged Warblers prefer less herb/shrub cover (Fig 2). Golden-winged Warblers prefer higher elevation sites with a smaller slope (Fig 2). Notably, Audubon and eBird models had similar variable importance values and shapes of response curves (Table 1, Fig 2). These findings demonstrate the importance of considering land use variables at different spatial scales when studying Golden-winged Warbler habitat, since different variables are important at different scales and variables may have

opposite effects depending on scale. In the following discussion of our results, we outline the similarities and differences between our models and the Golden-winged Warbler Working Group (GWWG) Appalachian Region management guidelines, provide management recommendations based on our results, and conclude with further applications of our work.

While Audubon and eBird models were highly similar, there were a few notable differences that can likely be attributed to either differences in sample size or differences in data collection methods. First, several variables contributed to the Audubon model that did not contribute to the eBird model, including forest of height 25-50m at the landscape scale, developed land at the landscape scale, and aspect. Second, nearly all important variables showed a stronger response or a greater baseline probability of occurrence in the Audubon model compared to the eBird model. Importantly, with a sample size of N = 279, the Audubon dataset had over 3x more presence points than the eBird dataset (N = 86). With a sample size an order of magnitude larger, the Audubon model may have been able to pull out patterns in the data that were not as strong in the eBird dataset. Our model tuning process led us to select a regularization multiplier of 15, which is relatively high (default is 1). The regularization process in Maxent protects against overfitting by applying a penalty to each term that is included in the model [58, 61, 66]. Regularization limits the size of variable coefficients, which is the most likely explanation for why terms in the Audubon model show stronger responses. Terms with high penalties may be completely removed from the model, resulting in a variable importance of 0 and a flat response curve, which is the most likely explanation for the variables that do not contribute to the eBird model but do contribute to the Audubon model. Thus, patterns seen in the Audubon dataset may be present but less detectable in the eBird dataset due to the smaller sample size, and our large regularization multiplier eliminated these variables from the eBird model to prevent overfitting. Interestingly, the Audubon model was better at predicting eBird presence values (AUC = 0.81) than the eBird model was at predicting Audubon presence values (AUC = 0.72), which is likely also due to the difference in sample size between the two datasets.

Alternatively, it is also possible that the differences in data collection methods led to the dissimilarities we see between Audubon and eBird models. For instance, it is possible that Audubon surveys are deliberately conducted away from developed areas based on management guidelines, while eBird surveys are biased towards more populated areas, resulting in a negative association between probability of occurrence and development in the Audubon model and no effect of development in the eBird model. Based on the relatively high variable importance values of the developed land variable in the Audubon model, we believe that this difference is likely due to both sample size and data collection methods (Table 1). Young forest at the local scale had a much smaller effect in the eBird model compared to the Audubon model (Fig 2). Audubon observers may be searching for Golden-winged Warblers in early successional habitat that is more structurally complex (mix of young forest, herb, and shrub), while eBird observers may be recording birds in less complex habitat with less young forest. However, unlike the developed land variable where variable importance values were very different between Audubon and eBird models, young forest at the local scale contributed little to both models, suggesting that sample size alone may be driving this difference (Table 1). Notably, both mature forest at the landscape scale and aspect contributed very little to the Audubon model, and were eliminated from the eBird model, suggesting that these variables are not important predictors of Golden-winged Warbler presence in our study area (Table 1, Fig 2).

Our results are mostly compatible with GWWG management guidelines for the Appalachian Region with a few caveats. GWWG management guidelines call for >70% forest cover within 2.4km of a habitat patch and 60–80% forest cover within 240m of a habitat patch [26]. As discussed above, our results suggest that in Western North Carolina, mature forest cover at

a landscape scale is not an important predictor of suitable Golden-winged Warbler habitat (Table 1, Fig 2). Most likely, this result is due to the landscape being dominated by mature forest cover, so percent mature forest cover is not important in distinguishing background points from presence points. The GWWG recommends 15–55% herb/shrub cover within 240m of a habitat patch [26]. Our results support this recommendation, since herb/shrub cover at the local scale was positively associated with presence in both models (Fig 2). However, our models show that the lack of mature forest at the local scale is a more important predictor of warbler occurrence than the presence of herb/shrub cover (Table 1). Additionally, within 150m, presence sites on average had only 10% herb/shrub cover, which is much lower than the recommended cover within the larger buffer of 240m (S2 Table). This is likely due to the nature of landcover data to underrepresent early successional habitat but could also reflect the lack of available early successional habitat in our study area, which may force birds to use suboptimal habitats. GWWG management guidelines suggest maintaining 30–70% shrub and sapling cover within a habitat patch [26], which aligns well with our models that show a positive association between presence and percent young forest at a local scale (Fig 2). Finally, GWWG management guidelines indicate that developed land is unsuitable for breeding Golden-winged Warblers [26]. Road cover at the local scale is positively associated with presence (Fig 2), but this can be attributed to 1) the propensity for early-successional habitat to be near roads; and 2) surveyor bias due to accessibility. Developed land at the landscape scale is negatively associated with presence in the Audubon model from values 0–0.02, after which probability of occurrence is 0, indicating that Golden-winged Warblers are selecting habitat within a less developed landscape, which is congruent with GWWG management guidelines (Fig 2).

It is important to note that the GWWG outlines management goals to create ideal Golden-winged Warbler breeding habitat. In reality, Golden-winged Warblers may be selecting sites that are less than optimal based on what is available. For example, the lack of early successional habitat in our study area may force Golden-winged Warblers to use very small corridors of early successional habitat with unmeasurable (with spatial data) amounts of young forest or herb/shrub cover. Thus, differences between our model results and the GWWG recommendations do not disqualify those recommendations, but rather describe how habitat is being used in Western North Carolina in contrast to those recommendations.

Based on our results, we make the following management and conservation recommendations for the Western North Carolina subregion. Both the Audubon and eBird models identified elevation and mature forest within 150m as the most important predictors of Golden-winged Warbler presence (Table 1). This suggests that in Western North Carolina, elevation and mature forest at the local scale are driving the difference between background sites and Golden-winged Warbler breeding sites. We recommend that future Golden-winged Warbler survey and management efforts be concentrated on areas of the landscape at high elevations (>800m, based on Fig 2, see also S2 Table). Both models found lack of mature forest to be a more important predictor of Golden-winged Warbler presence than herb/shrub cover at the local scale (Table 1, Fig 2). In our study area, herb/shrub communities not mixed with trees are likely maintained through heavy human disturbance. Thus, Golden-winged Warblers are likely selecting early successional habitat with complex vegetation layers including herbaceous, shrub, and trees, and with relatively low human disturbance, which is consistent with the literature [10–15]. We recommend that local-scale habitat (150m) be maintained with structural complexity such that young trees are present but space between trees and open canopy allow herb/shrub communities to coexist with young forest. We recommend that survey efforts to locate previously unknown Golden-winged Warbler territories focus on smaller or more structurally complex patches of early successional habitat, which are likely more common and are more utilized by birds in the Western North Carolina region.

Both models had AUC values of $\geq 0.80$, indicating that the models performed well but there were discrepancies in the data that could not be explained by our predictor variables. Much of this variance can likely be explained by (1) the coarseness and quality of raster data and (2) the rareness of Golden-winged Warblers across our study area. Landcover data is an important tool used frequently in spatial ecology, but it has notable shortcomings, including coarse pixel size, limited ground-truthing, and limited ability to describe heterogeneous landscapes. Additionally, raster datasets provide a snapshot of a landscape in time and seasonal variation, along with ecological succession, complicates the ability of raster datasets to fully describe a habitat. The scarcity of Golden-winged Warblers presents a challenge when studying breeding habitat associations since their low abundance can be due to a variety of factors not related to availability of breeding habitat, including dispersal effects, availability and quality of wintering habitat, and migration routes and availability of stopover sites. All these factors can affect the abundance and spatial distribution of Golden-winged Warblers across the landscape of our study area. While our model predictions suggest low habitat suitability across our study area, there could be other factors contributing to the density and distribution of the species in Western North Carolina, and these unknown factors could help explain discrepancies in the data that are not explained by the models.

Notably, presence locations are not confirmed breeding attempts in either dataset. We infer that birds observed during the breeding months are using those habitats for breeding, but this comes with some degree of uncertainty. Since Audubon NC data were collected with the use of conspecific playback, which elicits a territorial response from breeding males, we are slightly more confident that Audubon presence points represent breeding territories (compared to eBird points), but neither dataset represents confirmed breeding data. To confirm breeding, breeding behavior such as copulation, sitting on a nest, or feeding young must be documented. Recently, eBird has promoted the use of Breeding Bird Atlas codes in eBird checklists, where observers may indicate whether they witnessed breeding behaviors. We strongly recommend that eBird users integrate the practice of documenting breeding behavior into their observations, as this would improve breeding data quality and the research products that use eBird data. With the recent initiation of the North Carolina Bird Atlas (ebird.org/atlasnc/home), which is using eBird as a data submission platform, more of these breeding behavior data will likely be available for North Carolina birds in the future.

Future research on the habitat associations of Golden-winged Warblers should focus on analysis of multiple sources of spatial data, perhaps of finer spatial resolution, at multiple spatial scales. Since early successional habitat transforms over time into mature forest in our study area, future models should incorporate environmental variables with high temporal resolution. Additionally, future field research should investigate the relationships between habitat variables and Golden-winged Warbler survival and reproductive success, in order to understand how populations will respond to land use and habitat changes. Our results show that eBird data can produce Maxent species distribution modeling results that are similar to results obtained from the more standardized Audubon NC survey data. Researchers should continue to utilize eBird data to answer ecological questions since eBird data tends to be more comprehensive across both space and time than other methods of data collection. Additionally, since structured surveys such as Audubon North Carolina surveys require time and financial resources and may create a higher level of disturbance from playback, eBird data should be considered as a viable alternative to traditional surveys when appropriate. However, increased detection probability with playback and the increased sample size of occurrences as a result underscore the importance of continuing to use structured survey protocols such as those used in the Golden-winged Warbler Atlas Project when necessary. At the least, eBird and more

traditional survey methods should be used in combination or to supplement each other to improve our understanding of Golden-winged Warbler distribution.

## Supporting information

**S1 Table. Data sources for landcover and topographic variables included in Golden-winged Warbler Maxent habitat distribution models.**
(PDF)

**S2 Table. Ecological niche description including mean values (± standard deviation) for all variables included in Maxent models.**
(PDF)

**S1 Dataset. Data extracted at Audubon NC locations used in Maxent modeling.**
(CSV)

**S2 Dataset. Data extracted at eBird locations used in Maxent modeling.**
(CSV)

**S3 Dataset. Data extracted at background locations used in Maxent modeling.**
(CSV)

## Acknowledgments

We thank Audubon North Carolina and Curtis Smalling for collecting and sharing Golden-winged Warbler Atlas Project data with us. We thank Highlands Biological Station (Highlands, North Carolina, USA) and James T. Costa for connecting Audubon North Carolina (and Aimee Tomcho) with Lauren Whitenack through the UNC-Chapel Hill Institute for the Environment Highlands Field Site student internship program. Lauren Whitenack and Allen Hurlbert conceived the idea for this paper and designed the experiment. Lauren Whitenack analyzed the data in R and wrote the manuscript. Sara Snell Taylor provided mentorship throughout the project, helped design the methods, and helped create figures. Aimee Tomcho contributed valuable subject matter expertise. All coauthors helped edit and revise the manuscript. Data were collected by Audubon North Carolina employees and volunteers as well as eBird contributors. We appreciate the two anonymous reviewers who provided detailed, helpful feedback that greatly improved this paper.

## Author Contributions

**Conceptualization:** Lauren E. Whitenack, Allen H. Hurlbert.

**Data curation:** Lauren E. Whitenack.

**Formal analysis:** Lauren E. Whitenack, Sara J. Snell Taylor.

**Investigation:** Lauren E. Whitenack, Sara J. Snell Taylor.

**Methodology:** Lauren E. Whitenack, Sara J. Snell Taylor, Allen H. Hurlbert.

**Project administration:** Lauren E. Whitenack, Allen H. Hurlbert.

**Resources:** Allen H. Hurlbert.

**Writing – original draft:** Lauren E. Whitenack.

**Writing – review & editing:** Lauren E. Whitenack, Aimee Tomcho, Allen H. Hurlbert.

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
