## [Decision Letter · Decision Letter 0]

19 Oct 2022

PONE-D-22-25984Comparing multiscale, presence-only habitat suitability models created with structured survey data and community science data for a rare warbler species at the southern range marginPLOS ONE

Dear Dr. Whitenack,

Thank you for submitting your manuscript to PLOS ONE. After careful consideration, we feel that it has merit but does not fully meet PLOS ONE’s publication criteria as it currently stands. Therefore, we invite you to submit a revised version of the manuscript that addresses the points raised during the review process.

We look forward to receiving your revised manuscript.

Kind regards,

Travis Longcore, Ph.D.

Academic Editor

PLOS ONE

Journal Requirements:

Additional Editor Comments:

The reviewers raise methodological issues that would need to be thoroughly addressed in a revision. Machine learning approaches, which might include random forest models in addition to the suggested Maxent approach may be warranted with these data and approach. Availability of the underlying data in accordance with PLoS ONE policy may not be covered by the current statement (should be ebird.org even though ebird.com redirects) but we can address that in concert with a revision.

Reviewers' comments:

Reviewer's Responses to Questions

**Comments to the Author**

1. Is the manuscript technically sound, and do the data support the conclusions?

Reviewer #1: Partly

Reviewer #2: Yes

2. Has the statistical analysis been performed appropriately and rigorously? 

Reviewer #1: No

Reviewer #2: Yes

3. Have the authors made all data underlying the findings in their manuscript fully available?

Reviewer #1: No

Reviewer #2: No

4. Is the manuscript presented in an intelligible fashion and written in standard English?

Reviewer #1: Yes

Reviewer #2: Yes

5. Review Comments to the Author

Reviewer #1: The topic of this research can be found interesting in the bird conservation community as it targets a rare warbler species. Authors used 2 different datasets to compare their ability to predict habitat suitability of the species. The manuscript needs to be revised in both aim and goals and methods. My main comment is about using GLM for presence-background points. It is showed by various studies that machine learning methods specifically Maxent are much better than GLM for presence background points. Authors might find following comments useful for their manuscript.

Line 71: In which regions these were conducted? I agree that species in different regions might have different habitat requirements. Please explain a little more this concept based on, for instance, evolutionary processes to justify this gap of knowledge. Also, there are some other warblers showing multi-scale habitat selection than can be mentioned here. For example, please see: Amirkhiz et al. Investigating niches and distribution of a rare species in a hierarchical framework: Virginia’s Warbler (Leiothlypis virginiae) at its northeastern range limit. Landscape Ecol 36, 1039–1054 (2021). https://doi.org/10.1007/s10980-021-01217-7

Line 91. It was difficult for me to understand how studying habitat associations and genetic studies are related. This knowledge gap needs to be explained better to justify this study.

Lines 107-118: If this is a knowledge gap which this study is based upon, authors need to compare current survey area to what they found as “potential suitable habitats” and then recommend explicit management actions.

Lines 122-127: This is not really what ebird data are. have structured data gathered based on specific protocols to meet specific goals and standards. Please review this reference: https://cornelllabofornithology.github.io/ebird-best-practices/

Lines 1330-135: This justification needs to be reconsidered. Ebird data are structured data and heavily wetted by experts. They are not just some points. Please see the above reference.

Line 140. Authors mention in the introduction that the lack of knowledge on differences between habitat associations of this species in their study area and other areas is a justification for conducting their study so their goal needs to be accordingly.

Line 147: please explain what this focal area is

Line 150: Please add state lines to the U.S map. Also add breeding range and the focal area mentioned above

Line 153: One of the main benefits of ebird data is having absence points. Why did not authors use them in their study? Also, ebird data has strong standards to reduce the impact of sampling bias. Did authors follow ebird standard process?

It seems Audobon NC and ebird data have been gathered under very similar conditions. One of the goals of this study is to compare these 2 datasets. Thus, it is necessary to explain, in introduction, why this is an important research question? What are differences and how these differences can affect ecological studies or management actions. These can be explained as hypotheses or research questions.

Line 166: why 100 meters?

Line 173: How about climatic and topographic variables? If this study is all about associations with landcover data, goals and objectives should be restricted accordingly, assuming other habitat factors are constant or have no associations. Also, using only 2014 LC data for a dataset covering 200-2020 is based on the assumption that landcover did not change during this period. However, authors, as a reason for conducting this research, mentioned in introduction that habitat loss and human development are 2 main reasons for reductions in this species population. I would suggest either revising goals and objectives or using NLCD data which has a finer temporal resolution. If the latter, please use the closest NLCD layer for each year and extract corresponding landcover data for each point. Also, please check cropland data layers and https://www.ntsg.umt.edu/project/landsat/landsat-landcover.php . Vegetation height of land fire data still can be used along with NLCD or other landcover datasets.

Line 178: Which package? Citation. This comment applies to all methods and techniques.

Lines 180-185: based on what assumptions these buffers and measures were selected?

Line 186: how many presence points for each dataset?

Line 189: There are many papers proving that Maxent or other Machine learning methods have better performance for presence-only data. Merrow et al 2014 do not recommend using GLM for modeling habitat associations. They provide a range of options based on goals and the nature of data. Based on Merrow and many other papers I believe Maxent, or any other machine learning methods are better fit for these data. The main reason is using only presence data. I would consider GLM as an appropriate method if authors used absence data as well. please see the following reference:

Guisan, A., Thuiller, W., & Zimmermann, N. (2017). Habitat Suitability and Distribution Models: With Applications in R (Ecology, Biodiversity and Conservation). Cambridge: Cambridge University Press. doi:10.1017/9781139028271

Also, if investigating habitat associations is the main goal of this study, creating and interpretation of response curves should be a part of the paper. So, I strongly recommend including response curves in this study.

Please use more metrics for evaluating models if comparing 2 different datasets is the goal. Each metric has advantages and limitations. Please see Guisan et al 2017.

Reviewer #2: Lines 37 - 40: The second goal of the paper is not clear in the abstract (to determine if community science data produces similar distribution models as systematic sampling data).

Line 47. I suggest: Additionally “our results help to validate the use of bird data, since they produce similar species distribution modeling results…”

Methods

Method needs to specify the M used to model and if it was the same for eBird and Audubon NC data. Since it is well known that administrative divisions are not a good option. I suppose you used an M based on ecological characteristics relevant to the species.

You clearly state USFS land-fire raster resolution, however, It is not clear if EVH and NLCD rasters were already at 30m x 30m pixel size or if the resolution was changed.

Method also needs to specify how many of the presence points were used for training models and testing models in each case (eBird and Audubon NC data)

I think you should consider using ku.enm (Cobos et al. 2019) for the process, starting from model calibration. Among other benefits, ku.enm evaluates model performance using partial ROC, instead of area under the ROC curve, that has been prove to be a a more suitable indicator of statistical significance.

I wonder if there is information in eBird data to use only confirmed breeding presence, since I understand Audubon NC data are only confirmed breeding data. If I understood correctly a more thorough selection of eBird data could provide an even more similar model. If this is not the case you could at least discuss this in the corresponding section.

I understand the resolution you used somehow prevents you from using other environmental data such as temperature, however maybe you could have considered to use lidar data for topographic variables.

Even when you explain the importance of vegetation height for the species, I wonder if the use of a limited set of variables could be overestimating the importance of the variables when describing the niche.

I suggest to include a table describing the ecological niche. A table with means and SD of each variable for every model (Audubon and eBird)

The title is more focused on model comparison, however, the abstract and the discussion seem more focused on the importance of ENM proper description for conservation. I suggest to try to include both goals in the title.

6. PLOS authors have the option to publish the peer review history of their article (what does this mean?). If published, this will include your full peer review and any attached files.

Reviewer #1: No

Reviewer #2: No

---

## [Author Response · Author response to Decision Letter 0]

10 Jan 2023

Editor and reviewers, thank you for your detailed comments and suggestions. We believe your input has greatly improved our manuscript. We include responses to editor comments in our cover letter and responses to reviewer comments in our response to comments document.

---

## [Decision Letter · Decision Letter 1]

12 Mar 2023

Comparing multiscale, presence-only habitat suitability models created with structured survey data and community science data for a rare warbler species at the southern range margin

PONE-D-22-25984R1

Dear Dr. Whitenack,

We’re pleased to inform you that your manuscript has been judged scientifically suitable for publication and will be formally accepted for publication once it meets all outstanding technical requirements.

Kind regards,

Travis Longcore, Ph.D.

Academic Editor

PLOS ONE

Additional Editor Comments (optional):

Please note the need to provide data used in the modeling process during production.

Reviewers' comments:

Reviewer's Responses to Questions

**Comments to the Author**

1. If the authors have adequately addressed your comments raised in a previous round of review and you feel that this manuscript is now acceptable for publication, you may indicate that here to bypass the “Comments to the Author” section, enter your conflict of interest statement in the “Confidential to Editor” section, and submit your "Accept" recommendation.

Reviewer #1: All comments have been addressed

2. Is the manuscript technically sound, and do the data support the conclusions?

Reviewer #1: Yes

3. Has the statistical analysis been performed appropriately and rigorously? 

Reviewer #1: Yes

4. Have the authors made all data underlying the findings in their manuscript fully available?

Reviewer #1: No

5. Is the manuscript presented in an intelligible fashion and written in standard English?

Reviewer #1: Yes

6. Review Comments to the Author

Reviewer #1: I want to thank the authors for carefully considering all comments and suggestions. The manuscript is now fit to be published. Regarding data availability, authors can provide tables of their data used in the modeling process (The first column could be localities, and the rest can be corresponding extracted values of predictor variables.

7. PLOS authors have the option to publish the peer review history of their article (what does this mean?). If published, this will include your full peer review and any attached files.

Reviewer #1: No

---

## [Editor Report · Acceptance letter]

15 Mar 2023

PONE-D-22-25984R1 

Comparing multiscale, presence-only habitat suitability models created with structured survey data and community science data for a rare warbler species at the southern range margin 

Dear Dr. Whitenack:

I'm pleased to inform you that your manuscript has been deemed suitable for publication in PLOS ONE. Congratulations! Your manuscript is now with our production department. 

Kind regards, 

on behalf of

Dr. Travis Longcore 

Academic Editor

PLOS ONE